# In Situ Determination of Dry and Wet Snow Permittivity: Improving Equations for Low Frequency Radar Applications

Ryan W. Webb [1,2,*], Adrian Marziliano [1], Daniel McGrath [3], Randall Bonnell [3], Tate G. Meehan [4,5], Carrie Vuyovich [6] and Hans-Peter Marshall [4]

1   Center for Water and the Environment, University of New Mexico, Albuquerque, NM 87131, USA; amarziliano@unm.edu
2   Institute of Arctic and Alpine Research, University of Colorado Boulder, Boulder, CO 80309, USA
3   Department of Geosciences, Colorado State University, Fort Collins, CO 80523, USA; Daniel.McGrath@colostate.edu (D.M.); randall.bonnell@colostate.edu (R.B.)
4   Department of Geoscience, Boise State University, Boise, ID 83706, USA; tatemeehan@u.boisestate.edu (T.G.M.); hpmarshall@boisestate.edu (H.-P.M.)
5   US Army Cold Regions Research and Engineering Laboratory, Hanover, NH 03755, USA
6   NASA-Goddard Space Flight Center, Greenbelt, MD 20770, USA; carrie.m.vuyovich@nasa.gov
*   Correspondence: rwebb@unm.edu

**Abstract:** Extensive efforts have been made to observe the accumulation and melting of seasonal snow. However, making accurate observations of snow water equivalent (SWE) at global scales is challenging. Active radar systems show promise, provided the dielectric properties of the snowpack are accurately constrained. The dielectric constant (*k*) determines the velocity of a radar wave through snow, which is a critical component of time-of-flight radar techniques such as ground penetrating radar and interferometric synthetic aperture radar (InSAR). However, equations used to estimate *k* have been validated only for specific conditions with limited in situ validation for seasonal snow applications. The goal of this work was to further understand the dielectric permittivity of seasonal snow under both dry and wet conditions. We utilized extensive direct field observations of *k*, along with corresponding snow density and liquid water content (LWC) measurements. Data were collected in the Jemez Mountains, NM; Sandia Mountains, NM; Grand Mesa, CO; and Cameron Pass, CO from February 2020 to May 2021. We present empirical relationships based on 146 snow pits for dry snow conditions and 92 independent LWC observations in naturally melting snowpacks. Regression results had $r^2$ values of 0.57 and 0.37 for dry and wet snow conditions, respectively. Our results in dry snow showed large differences between our in situ observations and commonly applied equations. We attribute these differences to assumptions in the shape of the snow grains that may not hold true for seasonal snow applications. Different assumptions, and thus different equations, may be necessary for varying snowpack conditions in different climates, suggesting that further testing is necessary. When considering wet snow, large differences were found between commonly applied equations and our in situ measurements. Many previous equations assume a background (dry snow) *k* that we found to be inaccurate, as previously stated, and is the primary driver of resulting uncertainty. Our results suggest large errors in SWE (10–15%) or LWC (0.05–0.07 volumetric LWC) estimates based on current equations. The work presented here could prove useful for making accurate observations of changes in SWE using future InSAR opportunities such as NISAR and ROSE-L.

**Keywords:** snow permittivity; liquid water content; radar

## 1. Introduction

Snowmelt is the dominant freshwater resource for over a billion people globally [1,2] with recent studies showing its high monetary value [3]. Furthermore, seasonal snow is one of the fastest changing hydrologic states under current climate trends [4–6]. Due to the importance of snow and the rate it is changing, extensive efforts are being made

to observe the accumulation and melting of seasonal snow (e.g., [7]). Snow observation methods range from manual ground measurements to remote sensing techniques such as light detection and ranging (LiDAR) or active radar systems that rely on understanding the dielectric properties of the snowpack. However, making accurate observations of snow water equivalent (SWE) at the global scale is challenging.

Recent technological advancements have allowed limited mapping of SWE from remote sensing techniques. Passive microwave instruments such as the Multichannel Microwave Radiometer (SMMR), the Special Sensor Microwave Imager (SSM/I), and the Advanced Microwave Scanning Radiometer-EOS instrument (AMSR-E) have been used, though the poor spatial resolution of these datasets, sensitivity to varying snow properties, and inability to measure deep snow make these techniques less feasible for watershed-scale scientific applications [8–11], especially in complex mountainous terrain. One of the most successful methods in recent years has been the use of airborne LiDAR to obtain distributed snow depth maps and SWE estimates when density estimates are available [7,12,13]. Photogrammetry techniques using stereo satellite image pairs have also shown promise to produce snow depth maps [13,14]. However, LiDAR and photogrammetry techniques are generally limited to regional applications, require accurate and co-registered snow-off products, and are not viable during days of cloud-cover. Active microwave radar can overcome these limitations and has recently been used to provide estimates for changes in snow depth and SWE through differential interferometric synthetic aperture radar (In-SAR) [15–17]. Lower frequency InSAR techniques offer the potential to make observations of changes in SWE at high resolutions and are not hindered by cloud cover. However, InSAR techniques to observe SWE changes require a priori estimates of the dielectric properties of a snowpack. In particular, these low frequency active radar techniques are dependent on accurate estimates of the real part of the dielectric permittivity ($\varepsilon'$).

The $\varepsilon'$ determines the velocity of a radar wave through snow and has been used to investigate multiple regions of the cryosphere at spatial scales that range from the laboratory to multiple kilometers (e.g., [18–20]). Historically, $\varepsilon'$ has been used extensively in ground-penetrating radar (GPR) studies to estimate ice thickness [21], SWE [22], density [23], and snow liquid water content (LWC); [24]. Values of $\varepsilon'$ have primarily been validated for specific conditions such as polar firn and ice (e.g., [25]), with limited validation in seasonal snow. Additionally, $\varepsilon'$ for wet snow has been validated predominantly under idealized laboratory conditions due to the difficulty of making accurate in situ measurements of snow LWC. The studies that have made in situ observations of snow LWC have been limited to low values of liquid water (generally below 0.08 by volume) and the applied methods often sample different volumes of snow than those being measured for $\varepsilon'$ (e.g., [26–28]). Thus, there is a need for further evaluation of existing equations that quantify $\varepsilon'$ for seasonal snow under both dry and wet conditions, particularly with the forthcoming NASA-ISRO SAR Mission (NISAR) and Radar Observing System for Europe L-Band (ROSE-L) that have the potential to make global change in SWE products to make them feasible.

The goal of this work is to develop further understandings of the dielectric permittivity of seasonal snow under dry and wet conditions. We utilize extensive in situ observations of dielectric permittivity coincident with independent snow density ($\rho_s$) and LWC observations to complete the following objectives: (1) compare current permittivity equations for dry snow conditions against in situ observations, (2) compare current permittivity equations for wet snow conditions against in situ observations, and (3) determine if any improvements to current permittivity equations are necessary, and if so, recommend these improvements.

*Theoretical Background*

Many $\varepsilon'$ equations applied to snow are derived from the Polder and van Santeen [29] equation that assumes particles in mixed media are in the form of ellipsoids. The shape

of these ellipsoids may be described by the summation of their semi-axial ratios of the depolarization factors ($N_i$):

$$N_1 + N_2 + N_3 = 1, \tag{1}$$

where the subscripts signify the $N_i$ value for each respective semi-axis of the ellipsoid in three-dimensional space. When all $N_i$ values are equal (i.e., $N_1 = N_2 = N_3 = 0.33$), the particles are spherical (Figure 1a). Other $N_i$ value extremes describe discs (i.e., $N_i$ values of 0, 0.5, 0.5) and needles (i.e., $N_i$ values of 1, 0, 0; Figure 1a). When these ellipsoids have a known permittivity ($\varepsilon_k$) and are randomly oriented and distributed with a volumetric fraction ($\theta$) within a background material of a different permittivity ($\varepsilon_0$), the Polder and van Santeen [29] mixing formula for $\varepsilon'$ may be written as:

$$\varepsilon' = \varepsilon_0 + \frac{\theta}{3}(\varepsilon_k - \varepsilon_0) \sum_{i=1}^{3} \frac{\varepsilon'}{\varepsilon' + N_i(\varepsilon_k - \varepsilon')}, \tag{2}$$

In general, for dry snow, the shape of the snow crystals has a minimal impact on Equation (2) due to the relative similarities in permittivity between air and ice, with ice having a permittivity ~3 times that of air (Figure 1b). However, liquid water has a permittivity ~29 times that of ice resulting in a significant change in $\varepsilon'$ in the presence of even minimal amounts of liquid water (Figure 1c). The Polder and van Santeen [29] formula (Equation (2)) describes the theoretical basis for multiple equations predicting the dielectric behavior of snow and soil [30–32] that are regularly applied to seasonal snowpacks [13,33–35]. These equations are derived by assuming the shape of snow crystals for dry snow and liquid water inclusions for wet snow. Other equations used for commercially available sensors have also been developed empirically using laboratory techniques and a regression analysis (e.g., [36,37]). However, as previously mentioned, there has been limited testing of these equations for in situ snow samples, particularly for wet snow conditions. Because of this, estimates of LWC for wet snow conditions are often more sensitive to the equation chosen rather than intrinsic snowpack properties (Figure 1d). For further details on the permittivity models for mixed materials, Sihvola and Kong [38] and Di Paolo et al. [39] provide thorough reviews for dry snow conditions. For wet snow conditions, we provide a summary in terms of relative permittivity ($k$) as a function of snow density ($\rho_s$) and snow volumetric LWC ($\theta_w$) in Table 1. It is important to note that the equations shown in Table 1 are applicable at frequencies between 0.01 and 1.5 GHz where $\varepsilon'$ will not change significantly with frequency [18]. Equations used by instruments that are often utilized in the field are also included in Table 1. These instruments include the Snow Fork that uses Sihvola and Tiuri [31], a Denoth meter [30], an A2 Photonics WISe sensor [36], and the FPGA Company SLF Sensor [37]. Instruments often quantify $\varepsilon'$ of the observed media as $k$, defined as the ratio of $\varepsilon'$ for the observed media to the $\varepsilon'$ of a vacuum. Thus, we utilize $k$ for the purpose of consistency with our observations (described later in the Methods section).

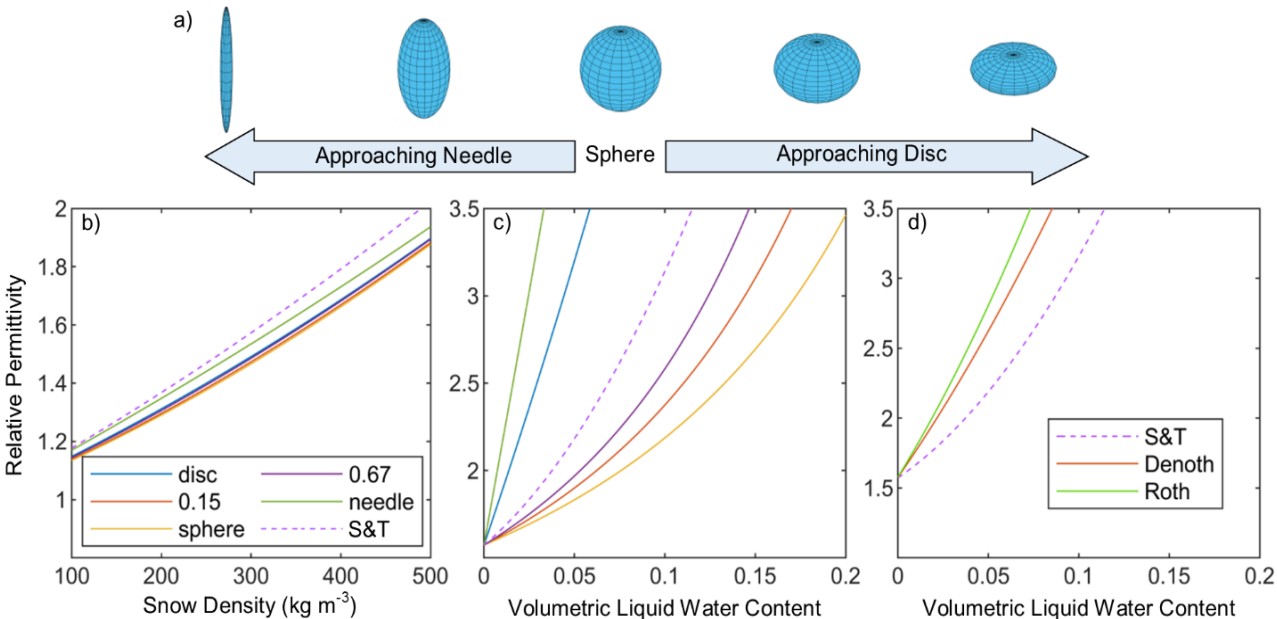

**Figure 1.** Examples of the Polder and van Santeen [29] formula for various shaped materials. (**a**) Diagrams of the three-dimensional ellipsoids defined by the depolarization factors ($N_i$). For the example plots, $N_1$ is noted in the legend and $N_2 = N_3$. (**b**) The Polder and van Santeen [29] model for the relative permittivity as a function of dry snow based on different $N_i$ assumptions and the Sihvola and Tiuri [31] equation (S and T) shown for comparison. (**c**) The Polder and van Santeen [29] model for relative permittivity as a function of volumetric liquid water content for snow with a density of 300 kg m$^{-3}$ based on different $N_i$ assumptions and the Sihvola and Tiuri [31] equation shown for comparison. (**d**) Comparison of Sihvola and Tiuri [31], Denoth et al. [26], and Roth et al. [32] equations for volumetric liquid water content of snow with a density of 300 kg m$^{-3}$.

**Table 1.** Common equations that relate snow density ($\rho_s$, kg m$^{-3}$) and volumetric liquid water content ($\theta_w$, cm cm$^{-3}$) to the relative permittivity ($k$) at low frequencies (0.01–1.5 GHz).

| Reference | $\theta_w$ Range (cm cm$^{-1}$) | Equation for Relative Permittivity, $k$ (Unless Otherwise Specified) |
|---|---|---|
| Sihvola and Tiuri [31] | 0.005–0.10 | $1 + 1.7\left(\frac{\rho_s}{1000} - \theta_w\right) + 0.7\left(\frac{\rho_s}{1000} - \theta_w\right)^2 + 8.7\theta_w + 0.007(100\cdot\theta_w)^2$ |
| Denoth [30] | 0.0–0.09 | $1 + 1.92\left(\frac{\rho_s}{1000}\right) + 0.44\left(\frac{\rho_s}{1000}\right)^2 + 18.7\theta_w + 45\cdot\theta_w^2$ |
| Roth [32] | - | $\left[9.38\cdot\theta_w + 1.78\cdot\frac{\left(\frac{\rho_s}{1000} - \theta_w\right)}{0.917} + \left(1 - \frac{\left(\frac{\rho_s}{1000} - \theta_w\right)}{0.917} - 100\cdot\theta_w\right)\right]^2$ |
| Kendra et al. [40] | 0.0–0.1 | $1 + 1.7(\rho_s - \theta_w) + 0.7(\rho_s - \theta_w)^2 + \Delta$ <br> $\Delta = 0.02(100\cdot\theta_w)^{1.015} + \frac{0.073(100\cdot\theta_w)^{1.31}}{1.0122}$ |
| Lundberg and Thunehed [41] | - | $(1 + 0.851\rho_s + 7.093\cdot\theta_w)^2$ |
| A2 Photonics [36] | 0.0–0.2 | $1 + 1.202\left(\frac{\rho_s}{1000} - \theta_w\right) + 0.983\left(\frac{\rho_s}{1000} - \theta_w\right)^2 + 21.3\cdot\theta_w$ |
| FPGA [37] | 0.0–0.2 | $\theta_w = \frac{0.271\left(k - k_{dry}\right)^3 - 2.688\left(k - k_{dry}\right)^2 + 10.337\left(k - k_{dry}\right)}{100}$ <br> $k_{dry} = -0.0083\left(3.44 \times 10^5 - 239.8(\rho_s - 100\cdot\theta_w)\right)^{0.5} + 4.893$ |

## 2. Materials and Methods

During the winter of 2020 and the springs of 2020 and 2021, robust data collection efforts were conducted as part of NASA SnowEx campaigns, designed to address the primary gaps in snow remote sensing, which included testing various radar remote sensing strategies. As part of these efforts, multiple snow pit observations included profiles of $\rho_s$, $k$, and $\theta_w$. The presented analyses are based on data from four sites (Figure 2). One site is Grand Mesa, CO where data from 146 snow pits that include $\rho_s$ and $k$ profiles collected

during a two-week observation period from 28 January 2020 to 12 February 2020 [42]. Further observations of $\rho_s$ and $k$ profiles were collected weekly in the Jemez Mountains, NM from 22 January 2020 to 4 March 2020 [43]. A third and fourth site for observations added measurements with a melt calorimeter for $\theta_w$ estimates in the Sandia Mountains, NM on a weekly basis from 25 February 2020 to 28 April 2020 [44] and Cameron Pass, CO from 9 March 2021 to 20 May 2021 [43]. For this study, Grand Mesa, CO and Jemez Mountains, NM data were used for dry snow conditions and the other two sites focused on data collection during wet snow conditions.

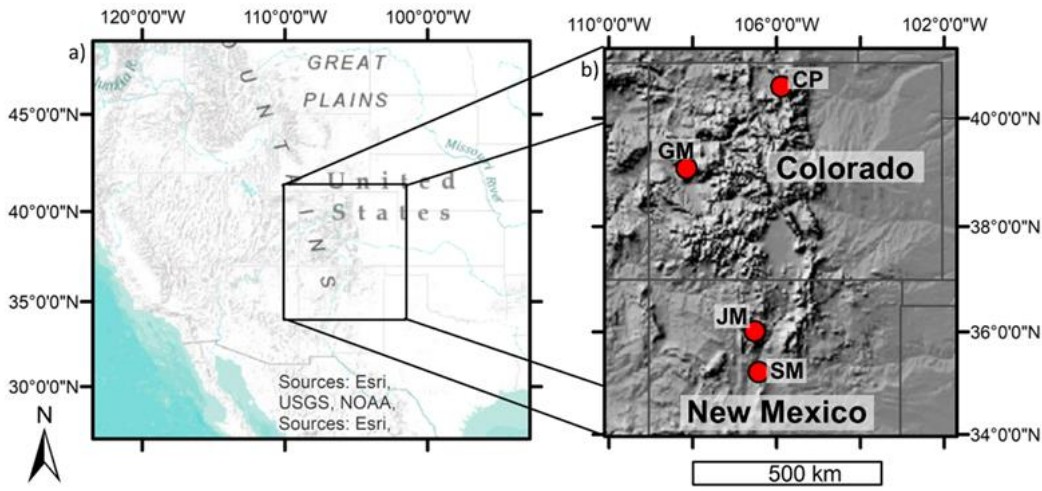

**Figure 2.** Site locations for Cameron Pass, CO (CP); Grand Mesa, CO (GM); Jemez Mountains, NM (JM); and Sandia Mountains, NM (SM) displaying (**a**) the relative location in the western U.S. and (**b**) a hillshade terrain map of Colorado and New Mexico.

Snow pit observations for all sites measured $\rho_s$, $k$, and snow temperature in 10 cm increments for the entire snowpack profile. Additionally, grain size, grain type, and manual hand wetness estimates were collected for each identified layer in the snowpack. Two profiles of $\rho_s$ were collected using a 1000 cm$^3$ wedge cutter and an electronic scale with 1 g precision to obtain observations in kg m$^{-3}$. Observations of $k$ were made using an A2Photonics WISe sensor that has a well-constrained measurement volume of 325 cm$^3$ (Figure 3a) and observations output as $k$ with precision to three decimal places. These $k$ values were independently verified using ground-penetrating radar estimates of $k$, using a combination of radar travel-time and manual depth measurements, for coincident snow pits (Appendix A). This independent verification resulted in mean absolute error (MAE) values of 0.106 in $k$. The error was found to be higher for wet snow conditions relative to dry snow with MAEs of 0.217 and 0.034, respectively (Figure A1). When melt calorimeter observations were made (described in further detail below), a ~25 g snow sample was taken directly from the WISe sensor. Snow temperature profiles were made using a dial stem thermometer with an accuracy of 1 °C. Grain size and type were made with a crystal card and hand lens. Hand wetness observations were made in accordance with the International Classification of Seasonal Snow on the Ground [45] that is commonly applied (e.g., [46]).

For the analyses in this study, we considered the snowpack to be dry when temperatures were below 0 °C and the hand wetness observations confirmed a "dry" estimate throughout the profile. When comparing dry $\rho_s$ to $k$, we used the mean $\rho_s$ and mean $k$ for each snow pit to account for random errors that may have occurred from the natural variability of $\rho_s$ and the differences in measurement volumes for each method described above; therefore, these estimates represent averages of more than 10 independent observations of dry $\rho_s$ and $k$. To compare the $k$ of wet snow to melt calorimeter estimates, we made direct comparisons since the calorimeter samples were taken directly from the measurement volume of the WISe sensor.

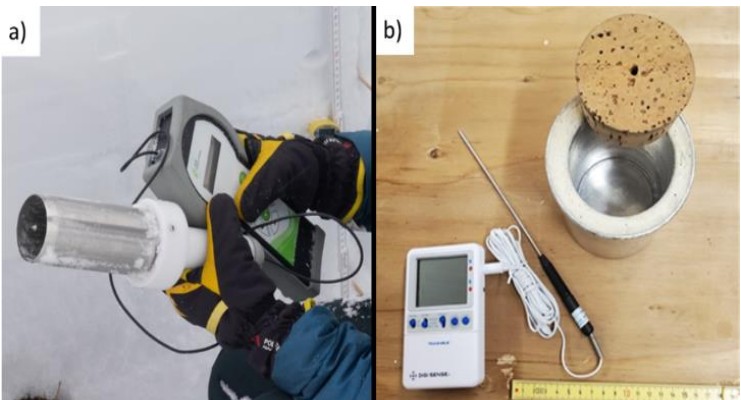

**Figure 3.** Image of (**a**) the A2 Photonics WISe Sensor and (**b**) the field calorimeter and thermometer used for this study.

The melt calorimeter is a double-walled insulated container with a 250 mL capacity (Figure 3b). Calorimeter measurements were made using a digital scale with 1 g precision and a thermometer with 0.01° precision and factory calibrated accuracy of 0.1 °C. Approximately 60–80 g of water with a temperature of 30–40 °C was added to the calorimeter and shaken to mix. The mass and temperature of the water in the calorimeter were then recorded. A snow sample (targeting 20–30 g) was quickly taken from the WISe sensor immediately after the $k$ measurement and placed in the calorimeter as quickly as possible. The calorimeter was then shaken to mix the water and sample for one minute to completely melt the snow sample. After mixing, the total mass of the water and melted snow sample was recorded along with the final temperature. The gravimetric LWC ($W$) was then calculated in a similar fashion to Kawashima et al. [27]:

$$W = 1 - \frac{C}{L}\left[\frac{M_w(T_w - T_F)}{M_s} - T_F\right],\tag{3}$$

where $C$ is the specific heat of water ($4.2 \times 10^3$ J kg$^{-1}$K$^{-1}$), $L$ is the latent heat of fusion for ice ($3.34 \times 10^5$ J kg$^{-1}$), $M_w$ is the mass of the warm water prior to the snow sample being added, $M_s$ is the mass of the snow sample, $T_w$ is the starting temperature of the warm water, and $T_F$ is the final temperature of the mixture. The $W$ values were then converted to $\theta_w$ by multiplying $W$ by the associated $\rho_s/1000$. Melt calorimeter observations were collected in the Sandia Mountains and Cameron Pass sites, including snow pits in open and forested conditions, after the snowpack became isothermal for a total of 92 independent wet snow observations.

## 3. Results

### 3.1. Dry Snow Observations

Snow pit observations [42,43] during the 2020 data collection resulted in a total of 149 snow pits for measurements of dry $\rho_s$ and associated $k$. These snow pits were from the Grand Mesa and Jemez sites. Snow pit depths ranged from 0.5 to 1.5 m. Mean dry $\rho_s$ ranged from 210 to 360 kg m$^{-3}$ and mean $k$ measurements ranged from 1.25 to 1.55 (Figure 4a). A regression analysis of $k$ as a function of dry $\rho_s$ resulted in an r$^2$ value of 0.57 and a root mean squared error (RMSE) of 0.03 with a standard deviation of 0.039 (Figure 4a). The equation for this regression of k as a function of dry $\rho_s$ is:

$$k = 1.0 + 0.0014\rho_s + 2 \times 10^{-7}\rho_s^2,\tag{4}$$

when compared with other equations such as the commonly used Sihvola and Tiuri [31], our in situ observations resulted in lower $k$ values and more similar to a less commonly applied Stein et al. [47] (Figure 4a).

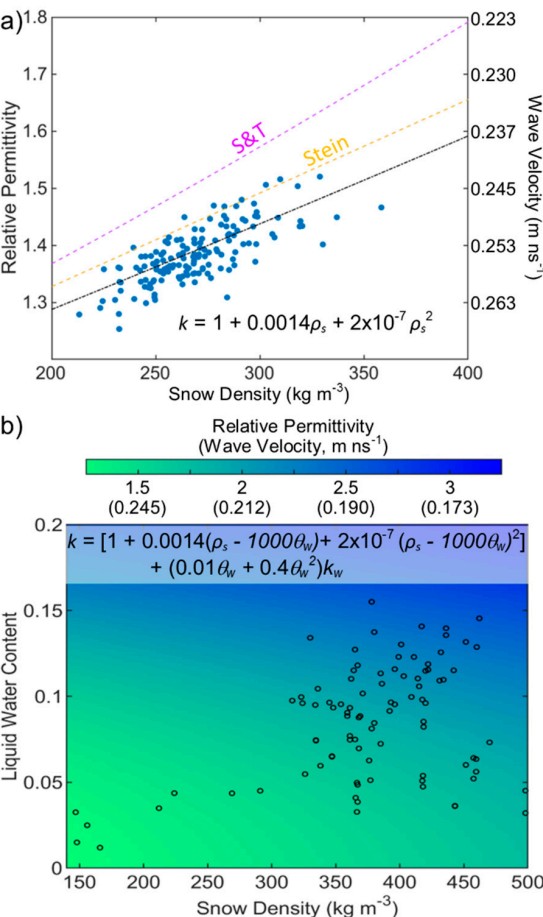

**Figure 4.** Results of in situ data comparisons for (**a**) dry snow with Sihvola and Tiuri [31] (S and T) and Stein et al. [47] shown for comparison to our regression line; and (**b**) for calorimeter-based liquid water content observations.

### 3.2. Wet Snow Observations

The Snow pit observations at the Sandia Mountains and Cameron Pass sites resulted in 92 observations with LWC present and isothermal conditions (necessary for appropriate application of Equation (3)). The values of $\rho_s$ ranged from 147–498 kg m$^{-3}$, observations of $\theta_w$ ranged from approximately 0.01 to 0.16, and $k$ measurements from 1.15 to 2.83. A regression analysis of $k$ as a function of $\rho_s$ and $\theta_w$ resulted in a r$^2$ value of 0.37 with a RMSE of 0.22 and a standard deviation of 0.21. In terms of deviations from $\theta_w$, this regression resulted in an RMSE of 0.030 and a standard deviation of 0.032 (Figure 4b). This regression equation for $k$ as a function of $\rho_s$ and $\theta_w$ is

$$k = \left[1.0 + 0.0014(\rho_s - 1000\theta_w) + 2 \times 10^{-7}(\rho_s - 1000\theta_w)^2\right] + \left(0.01\theta_w + 0.4\theta_w^2\right)k_w, \quad (5)$$

where $k_w$ is the relative permittivity of liquid water at 0 °C (~87.9) and the bracketed portion of the equation is the background effect of dry snow permittivity, described using Equation (4).

When compared to existing equations, Equation (5) aligns most closely with the SLF sensor equation for observations of $\theta_w$ greater than ~0.07 (Figure 5). However, for lower $\theta_w$ values Equation (5) aligns well with Sihvola and Tiuri [31]. For the snowpack conditions present at our sites, many of the equations, including four data points when our presented regression is used, resulted in a negative calculated $\theta_w$ that is physically unrealistic (Figure 5). These negative values were due to the actual background $k$ being lower than the assumed value in the $\theta_w$ equation (bracketed part of Equation (5)) and the effect of liquid water not being large enough to overcome this.

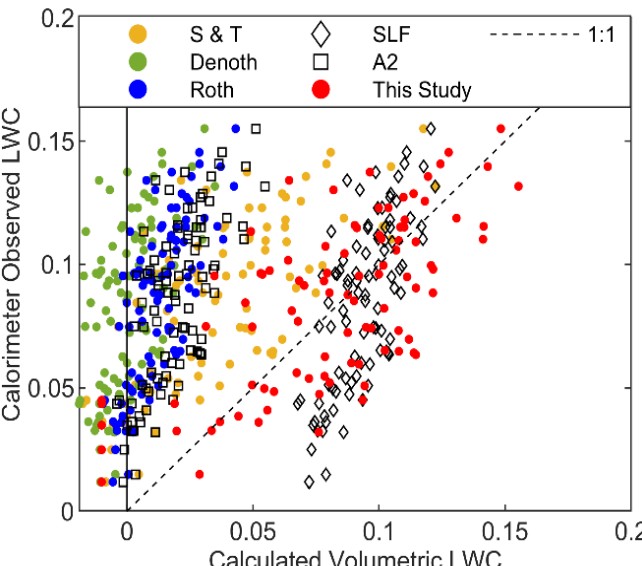

**Figure 5.** Comparisons of the calorimeter observed liquid water content and the calculated liquid water content based on the observed relative permittivity and density observations for common equations and the regression presented in this study.

## 4. Discussion

To our knowledge, this is the most robust in situ testing of permittivity equations for snow that has collected more datapoints than previous validation studies, particularly those investigating wet snow conditions. While sensors and equations have been compared in the past for in situ snow sampling, errors occurred as a result of differing sample volumes (e.g., [26,46]). In fact, *Denoth* et al. [26] mention that data were discarded due to the high variability during the comparison attempts. Furthermore, it is important to note that equations may result in different estimates of $\rho_s$ and $\theta_w$ for the same snowpack conditions as we have shown (Figures 4 and 5). This is likely a result of differing assumptions for $N_i$ in Equation (2) to theoretically derive some of the equations and/or variable conditions for laboratory tests used to empirically derive other equations.

For our dry snow tests, we showed reasonable agreement in the shape of the permittivity curve as a function of $\rho_s$. However, differences occurred between our in situ observations and previous equations (Figure 4a), particularly with one of the more commonly applied equations [31]. The revised equation presented here is most similar to Stein et al. [47], that takes a simple linear form. We explored a similar linear fit, but Equation (4) resulted in a slightly higher $r^2$ value (0.57 for Equation (4) and 0.55 for a linear fit). Additionally, it is important to note that our regression and data points were outside of the uncertainty bounds recently suggested by Di Paolo et al. [39]. Our analysis shows that different assumptions may hold more/less valid for varying snowpack conditions in different climates, though further testing is necessary. However, the relative similarities in $\varepsilon'$ between air and ice resulted in small differences between equations for dry snow conditions, particularly when compared to differences between equations in wet snow conditions (Figure 5). Our analysis highlights that care should be taken when applying an equation developed under conditions that differ from conditions for a site of interest. For example, for dry snow conditions common equations could underestimate snow density by ~50 kg m$^{-3}$ and the resulting SWE by 10–15% (Figure 4a) and estimates of $\theta_w$ could be underestimated by 0.05–0.07 (Figure 5). Common assumptions to consider when transferring equations from one condition to another may include factors such as snow climate that results in differing snow crystal size, form, and/or orientation, which influence the dielectric properties (Figure 1). It is important to note that the random orientation assumption in Equation (2) may not hold true in many continental snowpack conditions where metamorphism, and the resulting orientation, will predominantly occur

in the direction of the temperature gradient (i.e., vertically) resulting in a more organized crystal orientation [48], though further testing is necessary. These factors will also change the water retention properties of the snowpack and the resulting shape of water inclusions.

The shape of snow crystals and water inclusions is a likely reason for the differences in $\theta_w$ estimates between our in situ observations and previous equations. Previous equations and assumptions have shown to hold relatively true for lower values of $\theta_w$ within the pendular regime (i.e., liquid water is held in disconnected inclusions). However, as $\theta_w$ increases towards and into the funicular regime (i.e., liquid water is held within connected pathways) these shape assumptions break down. This is likely the result of preferential flowpaths that water follows in multiple directions as it percolates through the snowpack [49,50]. This is supported by our regression for wet snow conditions crossing over multiple $N_i$ shape curves using the Polder and Van Santeen [29] theoretical model (Figure 6). While it may be possible to solve for $N_i$ values that match Equation (5), it is unclear how physically representative of crystal shape and water inclusion this would be since snow metamorphism occurs quickly in the presence of liquid water and the shape of the water inclusion may change drastically from the pendular to funicular regimes. Furthermore, laboratory testing often involves sieving snow crystals to create ideal uniform conditions that do not occur in situ. However, these laboratory tests appear to compare reasonably well in the pendular regime with low liquid water contents (Figure 6a). With this comparison we can see that our in situ based regression followed a similar curve shape as previous equations, but with a shallower slope of the curve at low values of LWC. Analysis of the relative change in $k$ as a function of $\theta_w$ further supports our interpretation that the water inclusions change shape as they become more connected throughout the snowpack with increasing $\theta_w$. As $\theta_w$ increases towards 0.3 and above Equation (5), it approaches the curve of a sphere for Equation (2), though further testing is necessary for values of $\theta_w$ above 0.2.

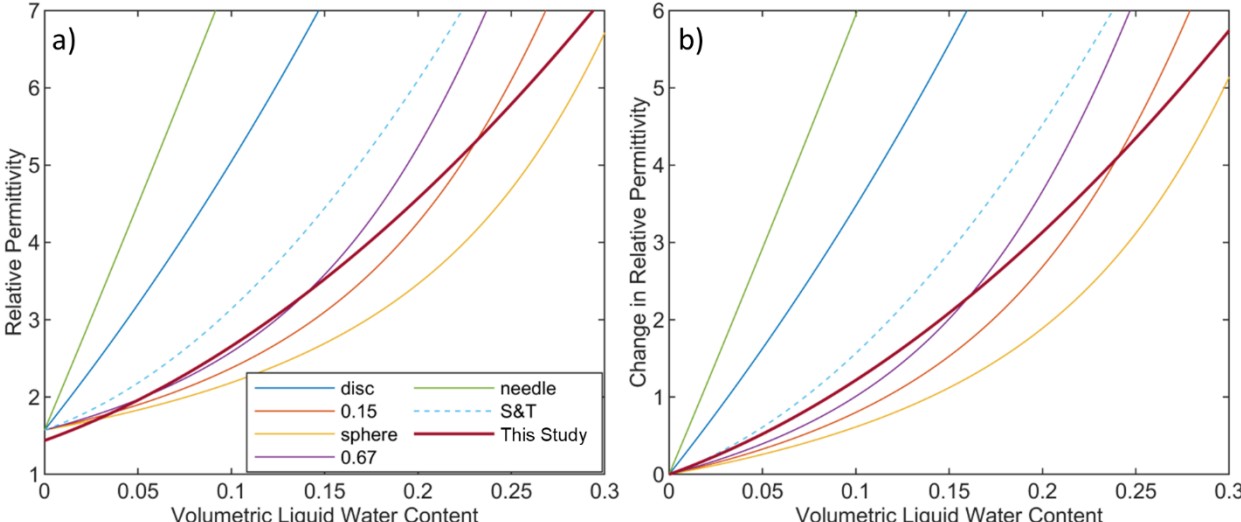

**Figure 6.** Comparison of relative permittivity equations based on the volumetric liquid water content of snow with a dry density of 300 kg m$^{-3}$. Equations shown are the Polder and van Santeen [29] formula for various shaped materials, Sihvola and Tiuri [31] (S and T), and the current study (This Study). Panel (**a**) shows the direct comparison of equations whereas panel (**b**) displays the relative change in permittivity for each equation as a function of the volumetric liquid water content.

*Uncertainty and Future Work*

When analyzing these $\theta_w$ data, it is also important to consider the uncertainty in our melt calorimeter observations. When using Equation (3), the precision and accuracy of our instrumentation resulted in a $\theta_w$ uncertainty estimate of approximately 0.017 [27]. However, due to the added potential error from transferring the snow sample from the WISe sensor to the calorimeter, we estimated an uncertainty of 0.02 $\theta_w$. We additionally conducted

multiple tests to determine energy losses by the calorimeter itself and found an average temperature change of ~0.2 °C per minute when a temperature gradient of ~15 °C was present between the warm water inside the calorimeter and the outside air temperature. This energy loss is easily accounted for and had minimal impacts on calculations using Equation (3). This total uncertainty is similar to other calorimeter studies (e.g., [27]) and may be reduced in the future by using a higher precision scale and improved insulation of the calorimeters. For the purposes of the present study, we consider this uncertainty reasonable due to the similarity with other studies and the regression RMSE of Equation (5) being of similar magnitude [18,24,30,31,37].

Though the $r^2$ value for our wet snow equation (Equation (5)) was relatively low at 0.37, it is worth noting that the uncertainty in the background $k$ from the dry snow equation, which has an $r^2$ of 0.57, factors into the spread. Thus, the change in permittivity as a result of liquid water increased the uncertainty by ~20%, whereas the dry snow portion of the equation (bracketed term of Equation (5)) had ~45% uncertainty. A more accurate understanding of the background permittivity could replace the bracketed term of Equation (5) to improve applications of this equation, though this may be difficult to acquire in the field as seasonal snowpacks often retain liquid water overnight once melt has begun [51,52]. Further investigations are necessary to reduce this uncertainty.

Future investigations may also consider expanding the range of $\rho_s$ and $\theta_w$ values for analysis. In this study, over 80% of samples of dry snow $\rho_s$ were between 240 and 300 kg m$^{-3}$ and, similarly, over 80% of samples for $\theta_w$ calculations were between 0.05 and 0.15 with $\rho_s$ over 300 kg m$^{-3}$. Future data collection efforts could target $\rho_s$ and $\theta_w$ values that were outside the sampling range of the current study. Additionally, these data were collected in continental snowpacks with snow depths less than 2.0 m and future data collection could further our understanding of the influence from varying snow climates and conditions to determine if different equations are necessary for differing snowpack regimes. Additionally, we recommend that future studies include additional parameters such as crystal structure and/or specific surface area in an attempt to reduce the uncertainty in empirical analyses.

However, our current results show that previous equations do not account for the changes in shape in relation to water held in snow pore spaces as increases in $\theta_w$ occur. This suggests that previous studies investigating the LWC of snow may have had errors in estimates as high as 0.1 $\theta_w$ depending on the choice of equation and the conditions present for the study (Figure 5). For conditions in the pendular regime, this error is the lowest whereas the funicular regime introduces the largest amount of error (Figures 5 and 6). For both dry and wet snow conditions, care should be taken when applying equations developed under different conditions. Future work that investigates the influence of varying snowpack conditions on the dielectric properties will assist in developing methods to utilize the forthcoming NISAR and ROSE-L satellite missions in producing accurate global changes in SWE products using low frequency microwave InSAR.

## 5. Conclusions

Our robust data collection effort and analysis showed that large differences exist between common equations and our field observations. These differences were found for both dry and wet snow conditions, illustrating that site-specific conditions strongly influence the corresponding empirical relations. For the snow conditions observed in this study, continental snowpacks with depths less than 2 m and densities less than 500 kg m$^{-3}$, we recommend Equations (4) and (5). Future work that utilizes the dielectric properties of snow should consider the snow climate when choosing an equation to apply.

**Author Contributions:** Conceptualization, R.W.W., D.M. and H.-P.M.; methodology, R.W.W., A.M., D.M. and R.B.; formal analysis, R.W.W.; investigation, R.W.W., A.M., D.M., R.B., T.G.M., C.V. and H.-P.M.; writing—original draft preparation, R.W.W.; writing—review and editing, R.W.W., D.M., R.B., T.G.M., C.V. and H.-P.M. All authors have read and agreed to the published version of the manuscript.

**Funding:** This research was funded by NASA Award # 80NSSC20K0921, NASA THP award 80NSSC18K0877, and NSF EAR Award #1824152.

**Data Availability Statement:** Datasets are available through the SnowEx online section of the National Snow and Ice Data Center, https://nsidc.org/data/snowex/data_summaries (accessed on 16 August 2021), and CUAHSI Hydroshare for the Sandia Mountains data, https://www.hydroshare.org/resource/f9c65581416b4021a860d648688d5d54/ (accessed on 16 August 2021), and for the Jemez, NM and Cameron Pass datasets, https://www.hydroshare.org/resource/46800976a58143cbbad11b9cd31d9e82/ (accessed on 16 August 2021).

**Acknowledgments:** We would also like to recognize the numerous scientists that participated during the SnowEx campaign, without which this work would not have been possible.

**Conflicts of Interest:** The authors declare no conflict of interest.

## Appendix A. Validation of the A2 Photonics WISe Sensor

In order to independently validate the permittivity values observed using the A2 Photonics WISe sensor, we utilized permittivity estimates from coincident ground penetrating radar (GPR) surveys. A GPR pulse is an electromagnetic wave that travels through the snowpack and is reflected off changes in material properties such as density, with the strongest reflection often from the snow–soil interface. The GPR data were processed to estimate the two-way traveltime (*TWT*) of the radar wave through the snowpack. The velocity of the radar wave (*v*) was then calculated as:

$$v = \frac{d_s}{\left(\frac{TWT}{2}\right)}, \tag{A1}$$

where $d_s$ is the depth of snow. The relative permittivity (*k*) of snow can then be calculated from:

$$k = \left(\frac{c}{v}\right)^2, \tag{A2}$$

where *c* is the speed of light in a vacuum (~0.3 m/ns).

We estimated the bulk *k* value for snow pits using GPR *TWT* and observed $d_s$ that were compared to the mean snow pit *k* values recorded using the A2 Photonics WISe sensor. In total, we had coincident observations for 28 snow pits, 17 in dry snow conditions and 11 in wet snow conditions. For all data points, a mean absolute error (MAE) in *k* values of 0.106 was found (0.034 for dry snow and 0.217 for wet snow; Figure A1). We attribute the higher deviation under wet snow conditions to the high spatial variability of liquid water in snow that is known to occur and the difference in volumes of influence between the GPR and WISe sensor profiles.

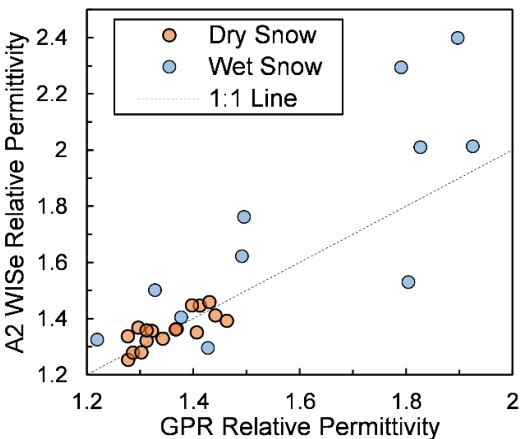

**Figure A1.** Comparison of relative permittivity estimates using a GPR and the A2 Photonics WISe sensor.

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
