# Peer review of "In Situ Determination of Dry and Wet Snow Permittivity: Improving Equations for Low Frequency Radar Applications"

_remotesensing, doi:10.3390/rs13224617_

Round 1
Reviewer 1 Report
The paper, In Situ Determination of Dry and Wet Snow Permittivity: Improving Equations for Low Frequency Radar Applications, is a robust and clearly written paper addressing the differences in equations used for dielectric properties of snow at different sites and environmental conditions. They highlight the lack of in situ testing of most equations, and now that they have brought this to the fore, the strong importance of site specific knowledge in order to select the most appropriate equation
The work has broad significance for remote sensing of snow in wet and dry snowpacks. The work discusses the importance of constraining the real part of the dielectric permittivity, in order to support more accurate estimation of wet snow and snow water equivalent, especially for upcoming insar missions. Added challenges come from most measurements being made under laboratory rather than field conditions.
Careful analysis of limitations of the work, uncertainty, and needed future work. This will be an important contribution to microwave remote sensing of snow properties.
I recommend publication.
Reviewer 2 Report
Review of remotesensing-1401513-peer-review-v1
Submission to Remote Sensing
Summary:
The authors measured the dielectric permittivity of dry and wet terrestrial snow along with measurements of snow density and liquid water content. In the paper they compare the density dependent dry snow permittivity to existing models, and the liquid water content dependent wet snow permittivity to existing models. They find a small difference between dry snow observations and previous models, with a new model from the authors’ observations resulting in a comparatively lower relative permittivity estimate. Variations in snow climate that results in differing snow properties is given as a contributing factor. The authors find a wide spread in relationships between observed and calculated liquid water content (LWC), in their Figure 5 which compares their new model for wet snow to previous ones. Here, the calculated LWC is based on models that relate observed relative permittivity and density observations to LWC. The authors attribute differences between their observations and previous LWC estimates to the shape of snow crystals and water inclusions, with higher LWC snow (i.e. funicular regime) creating more uncertainty.
The work is well justified in the need for more comparisons of in situ measurements to existing models, both in dry and wet snow cases, and more consideration for the possibility of regional variations in snow permittivity. Since knowledge of permittivity is important for radar based retrievals of snow properties, this paper should fit the defined scope for a Technical Note for Remote Sensing provided the major comments can be adequately addressed.
Major comments:
- There is considerable value in the in situ testing of permittivity equations and the authors have provided a large dataset of snow pit observations of snow density, relative permittivity, and LWC of dry and wet snow conditions for this purpose. This is certainly an interesting and insightful study but does not adequately achieve the stated objectives. The authors aren’t effectively testing current permittivity equations using their in situ observations and are instead deriving their own equations, unique to their own region and conditions, and comparing them qualitatively to pre-existing equations. The authors offer their own perspective on the confounding variables that create uncertainty in the existing equations, as well as their own, but do not provide any new quantitative information on these confounding variables in order to test and improve the permittivity equations. Testing these equations would require information on the shape, size, and orientation of snow grains in the dry snow case, data on the shape of snow grains and liquid water inclusions held in pore spaces in the wet snow case. Since it is not likely that these difficult measurements were made in detail during this study, the authors should restate their objectives and define their study more appropriately in the context of finding permittivity equations for the continental snow condition case encountered (and comparing them to previous equations but not testing those equations as such).
- In line with the above, the authors should include some data, e.g. photographs, if possible, to substantiate some of the assertions made about grain orientation, at least for the dry snow case. For example, lines 277-280 contain speculation about the assumption of random orientation not holding for the sampled continental snow conditions. If there are some snow pit wall photographs or snow layer structural analyses done, they should be used here.
- In reference [28], Tiuri et al., indicate that the dielectric constant of snow is almost independent of snow structure and therefore metamorphism. The authors somewhat contradict this with their own assertions about snow grain and water inclusions structures. The supporting literature and the contributions of instrument errors need to be carefully considered when making these assertions.
- The appropriate frequency ranges of equations should be stated where appropriate, e.g. Table 1. This is since the permittivity of wet snow is dependent on frequency (as well as volumetric water content). Similarly, if the work is motivated by the need to quantify the dielectric permittivity to take advantage of L-band missions (e.g. line 81), applicability to the L-band frequency needs to be detailed.
Specific comments:
(Line = L)
L25-26: Clearly state what the regression relationships are i.e. the variables.
L54: “… poor spatial resolution …”.
L83: Delete “Sentinel 12” since the rest describes “ROSE-L”.
L189: The error should be detailed here instead of in the discussion.
L305: Include “a)” in the figure.
Reviewer 3 Report
Review comments for “In Situ Determination of Dry and Wet Snow Permittivity: Improving Equations for Low Frequency Radar Applications” by Webb et al.
This manuscript describes in detail of the construction procedures of two new empirical relationships using observational data collected from the recent SnowEX campaign over four Colorado and New Mexico sites. The empirical relationships related measured snow permissivity (k) with snow density (rho_s) and liquid water content (theta_w) so to describe the dry snow and wet snow scenarios, respectively. It also compares the new empirical relationships against widely used ones, explained the reasons that potentially cause the discrepancies, and associated impacts to other applications. Uncertainties and their sources are discussed as well. At the end, future observations with wider range of theta_w and rho_s values from other areas on the Earth are recommended to complete these relationships and to avoid unrealistic extrapolation.
Overall I enjoy reading this paper and learning through it. My background is atmospheric hydrometeor remote sensing, where snow and ice surface contribution is always an “annoying contamination” to our signals. Therefore, works like SnowEX mission are highly appreciated by other communities, and the results and comparisons from this work is informative to us as well. I’m looking forward to a prompt publication of this work.
Some minor recommendations are below to help readers from different background than snow emissivity understand better of your work:
(1) I found the assumption of random orientation of snow particles on ground (Line 103) interesting. In hydrometeor remote sensing community, we’ve seen dominant horizontal orientation for stratiform snow aggregates globally. I’m wondering if the random versus horizontal assumption would significantly impact your “rho_s” estimation for your study.
(2) Equation 3: how to relate LWC (theta_w) with “W” parameter here?
(3) Line 218: any assertion on what the potential candidates are to cause lower k values?
(4) Line 327 – 329: here the uncertainty values are more or less “thrown out” to me without many explanations to relate to instrument accuracy? In-situ impact? I’m not sure if it’s partially explained more in the Appendix A. More detailed explanations or citations to related works are appreciated.
Round 2
Reviewer 2 Report
Thank you for providing a thorough response file outlining how the reviewer concerns were addressed and how the paper changed. The paper is much clearer and improved, and the technical note will be of interest and value to the snow and remote sensing community.